# Sediment controls dynamic behavior of a Cordilleran Ice Stream at the Last Glacial Maximum

Ellen A. Cowan [1]✉, Sarah D. Zellers [2], Juliane Müller [3,4], Maureen H. Walczak[5], Lindsay L. Worthington[6], Beth E. Caissie [7], Wesley A. Clary [6], John M. Jaeger [8], Sean P. S. Gulick [9], Jacob W. Pratt[1], Alan C. Mix[5] & Stewart J. Fallon [10]

The uncertain response of marine terminating outlet glaciers to climate change at time scales beyond short-term observation limits models of future sea level rise. At temperate tidewater margins, abundant subglacial meltwater forms morainal banks (marine shoals) or ice-contact deltas that reduce water depth, stabilizing grounding lines and slowing or reversing glacial retreat. Here we present a radiocarbon-dated record from Integrated Ocean Drilling Program (IODP) Site U1421 that tracks the terminus of the largest Alaskan Cordilleran Ice Sheet outlet glacier during Last Glacial Maximum climate transitions. Sedimentation rates, ice-rafted debris, and microfossil and biogeochemical proxies, show repeated abrupt collapses and slow advances typical of the tidewater glacier cycle observed in modern systems. When global sea level rise exceeded the local rate of bank building, the cycle of readvances stopped leading to irreversible retreat. These results support theory that suggests sediment dynamics can control tidewater terminus position on an open shelf under temperate conditions delaying climate-driven retreat.

[1] Department of Geological and Environmental Sciences, Appalachian State University, Box 32067Boone, NC 28608, USA. [2] School of Geoscience, Physics, and Safety, University of Central Missouri, Warrensburg, MO 64093, USA. [3] Alfred Wegener Institute, Helmholtz Centre for Polar and Marine Research, Am Alten Hafen 26, 27568 Bremerhaven, Germany. [4] Center for Marine Environmental Sciences, University of Bremen/MARUM, 28359 Bremen, Germany. [5] College of Earth, Ocean, and Atmospheric Sciences, Oregon State University, Corvallis, OR 97331, USA. [6] Department of Earth and Planetary Sciences, University of New Mexico, Albuquerque, NM 87131, USA. [7] Department of Geological and Atmospheric Sciences, Iowa State University, Ames, IA 50011, USA. [8] Department of Geological Sciences, University of Florida, Gainesville, FL 32611, USA. [9] Institute for Geophysics & Department of Geological Sciences, Jackson School of Geosciences, University of Texas at Austin, Austin, TX 78758, USA. [10] Research School of Earth Science, The Australian National University, Canberra, ACT 2601, Australia. ✉email: cowanea@appstate.edu

I n the present climate regime, Alaskan tidewater glaciers exhibit regionally asynchronous behavior with ~33% advancing[1] despite terrestrial glaciers being in retreat globally[2,3]. Similarly, in Greenland the stability of calving outlet glaciers is not uniform but is influenced by local conditions particular to their marine termini[4–6], as well as by thinning and acceleration attributed to climate change[7]. Both theory[8,9] and field studies[10,11] demonstrate that sediment yield from temperate glacial erosion and sediment transport can reduce water depth and restrict calving by building up morainal banks and ice-contact deltas. This behavior, which may last for centuries to millennia, comprises the advance phase of the tidewater glacier cycle (TGC) when calving is low while the morainal bank builds at the grounding line[3,12,13]. Iceberg flux may be eliminated if the bank rises above the sea surface and grows into an ice-contact delta. A delta or moraincal bank, however, eventually destabilizes when meltwater-controlled erosion creates an overdeepened, reverse sloping bed and a terminus grounded in progressively deeper water, initiating retreat[13]. Catastrophic collapse produces an ice-rafted debris mass accumulation rate (IRD MAR) peak due to increased calving rates lasting decades until it reaches a stable position in shallower water.

At the Last Glacial Maximum (LGM) (occurring from 26.5 to 19 ka cal BP[14]), the Cordilleran ice sheet discharged to the NE Pacific via ice streams through eight shelf-crossing sea valleys on the southern Alaskan margin[15–18]. Today, Bering Glacier is grounded on land, seaward of the St. Elias Mountains and Bagley Icefield and comprises 15% of the glacier ice in Alaska[19] (Fig. 1). Prior to the LGM, the Bering–Bagley Ice Stream (BIS) advanced to the shelf edge, carving a trough while adjacent areas on the shelf remained ice free[18,20]. Since the late Pleistocene, BIS has built a trough-mouth fan on the continental slope[17] and has been a major sediment source for the Surveyor Fan in the Gulf of Alaska[21].

Bounded by steep sidewalls, the morphology of the 25-km-wide, 55-km-long Bering Trough is similar to Alaskan fjords, where the TGC was first described (i.e., Taku Glacier[22] and Columbia Glacier[1]) except that it opens directly into the Gulf of

Alaska (Fig. 1). Sea level variability along the Alaskan coastline is largely a result of both active tectonics and isostasy[23], however estimates based on ice volume alone suggest ~134 m lowering during LGM[24]. Sea surface temperature (SST) reconstructions for the LGM[25,26], sedimentary architecture in the trough[16], and high accumulation rates of glacimarine mud[21,27,28] indicate that BIS existed under a temperate climate regime as a grounded tidewater ice cliff without a floating ice tongue or shelf[29].

Integrated Ocean Drilling Program (IODP) Site U1421 (59°30.4399′N, 143°2.7395′W) was drilled to 695.72 m CCSF-A (meters core composite depth below sea floor) at 721 m water depth on the continental slope[30]. This location is 10 km seaward of the BIS maximum advance at the shelf break and is directly in the path of icebergs and meltwater plumes but did not experience glacial erosion as on the shelf[18,31]. Integration of high-resolution multichannel seismic reflection data and U1421 chronology over ~130 ka shows that the upper 600 m was deposited in a glacial trough-mouth fan with sustained accumulation rates (from 0.5–1.0 cm year$^{-1}$)[18]. Here, we focus on the upper 116 m (Supplementary Fig. 1) because the 85.5% recovery of this part of the stratigraphic record affords the opportunity to develop a high-resolution reconstruction of dynamic ice stream behavior constrained by foraminiferal radiocarbon dates (Supplementary Fig. 2). Here we present evidence that within the BIS sediment dynamics at the Bering Glacier terminus have played a role similar to that of fjords over thousands of years during the LGM and subsequent deglacial. Our study demonstrates that the TGC operates in both open shelf and restricted fjord conditions in meltwater-dominated settings.

## Results
Between 114.4 and 98.7 m and between 46.6 and 38 m CCSF at Site U1421 no core was recovered in any of the three attempted holes (Fig. 2). Correlations among the core, logging data, and seismic profiles indicate that chaotic to semitransparent units with low-amplitude internal reflections represent these non-recovered intervals[32] (Supplementary Fig. 3). This is a seismic signature typically associated with downslope transport, such as by debris flows and turbidity currents that occurred on the upper slope when BIS terminus occupied the shelf break[16,18]. These sedimentary facies, occurring approximately between 26.3–26 and 18.2–17.9 ka cal BP, are difficult to recover by advanced piston coring from the D/V JOIDES Resolution and are interpreted as periods spanning several centuries when the BIS terminus was likely at the edge of the continental shelf. As discussed below, this interpretation is further supported by high adjacent sedimentation rates from a marine-terminating ice margin in an advanced position on the continental shelf (Fig. 2).

Recovered intervals with high sedimentation rates are associated with deposition of mud including varying amounts of sand and clasts ranging from granules to pebbles. Gradational contacts, random clast orientations, and the presence of well-preserved foraminifers and diatoms support deposition by rainout from the water column rather than downslope transport by gravity flows[16]. Recovered intervals with well-sorted muddy sediment represent periods when the grounding line retreated onto land and only fine-grained sediment was transported to Site U1421 (Figs. 2, 3, and 4d).

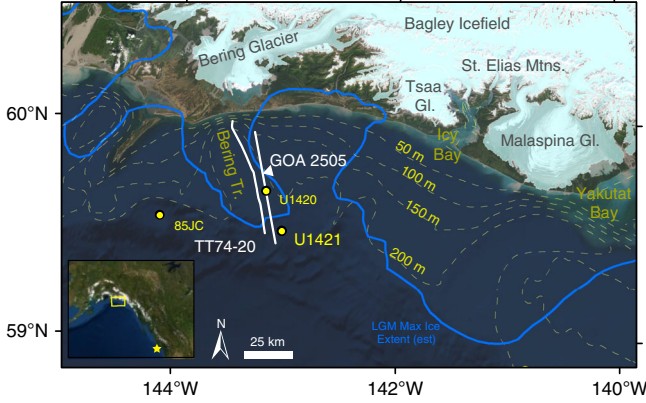

**Fig. 1 Study area.** Bathymetric map showing the shelf-crossing Bering Trough in southern Alaska. IODP Site U1421[30] is located on the continental slope, U1420[30] within the trough, and EW0408-85JC[27] to the west. The modern extent of Bering Glacier[76] is shown on the Landsat image, the estimated maximum Global LGM extent[20] is shown by the blue line. The location of seismic profiles GOA2505[66] and USGS TT74 Line 20[65] in Fig. 3 is shown. Start of line GOA2505 in Fig. 3 is indicated by white triangle. Location of MD02-2496[25] shown by star in inset. This map was created using resources available through Esri, DigitalGlobe, GeoEye, Earthstar Geographics, CNES/Airbus DS, USDA, USGS, AeroGRID, IGN, and the GIS User Community.

**Evidence for TGC.** The TGC is documented here using seismic reflection profiles and sedimentary facies analysis, coarse sand (250 μm–2 mm) derived IRD MAR, counts of diatoms, planktic and benthic foraminifera, and organic geochemical data (Figs. 2 and 5). High-resolution seismic reflection profiles on the shelf show high-amplitude, mappable reflections truncating older

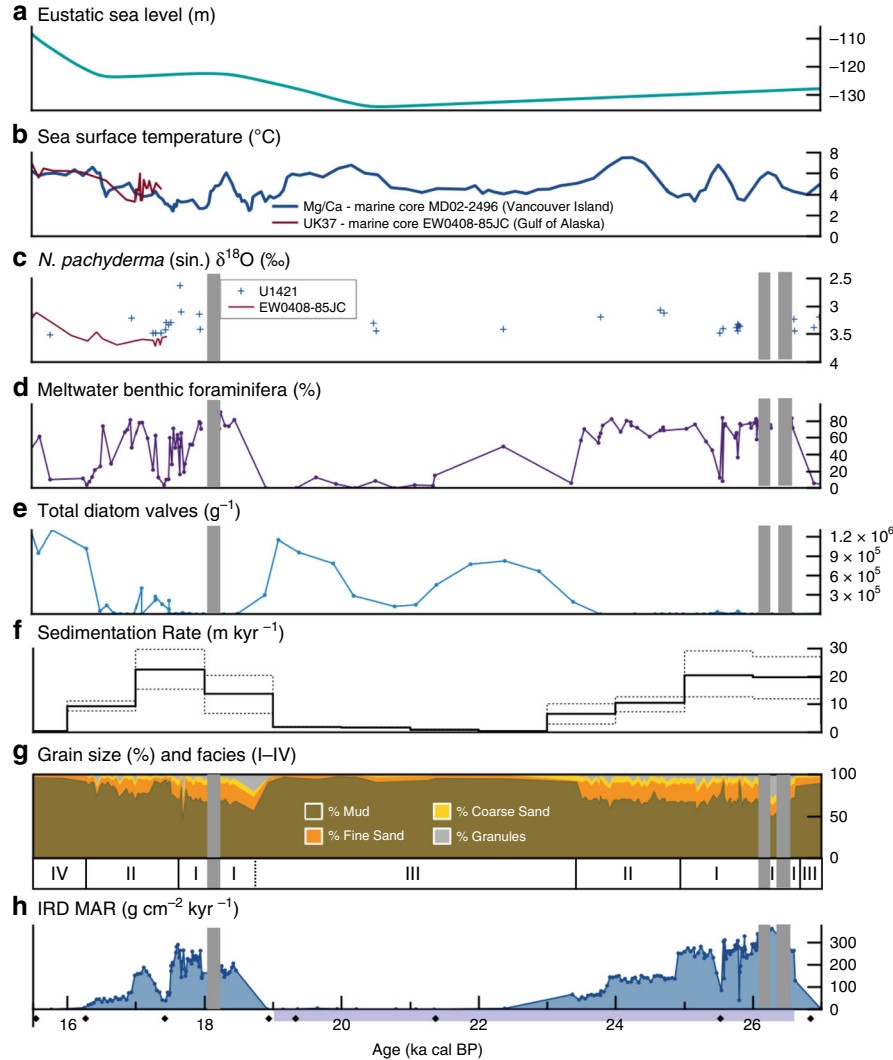

**Fig. 2 Multiproxy records from slope Site U1421 from 27 to 15 kyr ago. a** Global mean sea level curve (m)[24]. **b** Regional sea surface temperature proxies from EW0408-85JC[26] and MD02-2496[25] (Fig. 1). Mg/Ca values plotted using 4-pt moving average. **c** Values for δ[18]O analysis of *N. pachyderma* (sin.) in samples from site U1421 and survey site EW0408-85JC[27]. Error bars (±1 sigma) are smaller than symbol size. **d** Percentage of benthic foraminifera indicative of meltwater. **e** Diatom valves per gram sediment. **f** Mean sedimentation rates (m kyr$^{-1}$) binned at 1000-year intervals. One-sigma error is denoted by the dashed lines. **g** Distribution of grain sizes and lithofacies (I–IV). **h** Ice-rafted debris mass accumulation rate (g cm$^{-2}$kyr$^{-1}$). Gray shaded areas are intervals of no core recovery. Black diamonds are radiocarbon dates. Last Glacial Maximum from 26.5 to 19 ka cal BP[14] is shown by purple bar.

strata that we interpret as unconformities created by glacial erosion (advance). Semitransparent, low-amplitude to transparent facies overlying these surfaces represent sediment deposited during this retreat within the trough[18,31] (Fig. 3a). Mounded chaotic zones, sometimes with higher-amplitude, internal reflections, represent morainal banks and ice-contact deposits[16] (Fig. 3b at A–C). Despite poor recovery (<8%) at IODP Site U1420[30] on the shelf (Figs. 1 and 3a), the drilled rocks and washed pebbles that were recovered confirm the interpretation of reworking of shelf sediments by ice stream advance.

Facies I consists of a rapidly accumulating (~20 m kyr$^{-1}$) clast-rich diamicton or mud with abundant clasts with high-frequency peaks and troughs in IRD MAR averaging >240 g cm$^{-2}$ kyr$^{-1}$ (Fig. 2f–h). The repeated succession of peaks and troughs in IRD MAR record ice stream collapse, releasing high volumes of icebergs (high IRD) followed by slow advance of the terminus behind a moraimal bank (low IRD): these cycles are shown in detail in Supplementary Fig. 4. The typical pattern of rapid collapse and slower advance shown by the IRD MAR record is similar to observations of modern Alaskan glaciers[1,22] and model

predictions[13]. Facies I occurs between ~26.7 and 25.2 ka cal BP (cycles 1–6) and 18.7–17.6 ka cal BP (cycles 8–12) (Fig. 2g, h, Supplementary Fig. 4), and likely reflects a fluctuating tidewater BIS near the seaward end of the trough (Figs. 3 and 4). Likely positions of the outer and inner moraimal banks can be seen in the high-resolution minisparker profile shown in Fig. 3b.

Facies II has reduced sedimentation rates (~10 m kyr$^{-1}$) compared with Facies I, and consists of clast-rich diamicton and mud with dispersed clasts with average IRD MAR of ~100 g cm$^{-2}$ kyr$^{-1}$ (Fig. 2f–h). Facies II deposition is found from 25.2 to 23.4 ka cal BP and 17.6 to 16.3 ka cal BP, and records sustained retreat within the trough (Fig. 4, Supplementary Fig. 4). Tidewater glaciers are known to be sensitive to warmer SSTs and resulting near-terminus ocean circulation[33] and SST records from coastal British Columbia[25] and (corroborated in the Gulf of Alaska[26] for the younger interval) indicate warming of the NE Pacific coeval to the deposition of the Facies II units at Site U1421 (Fig. 2b). It is possible that warmer ocean temperatures induced calving at the terminus leading to the deposition of Facies II; an effect potentially enhanced if the moraimal bank was absent.

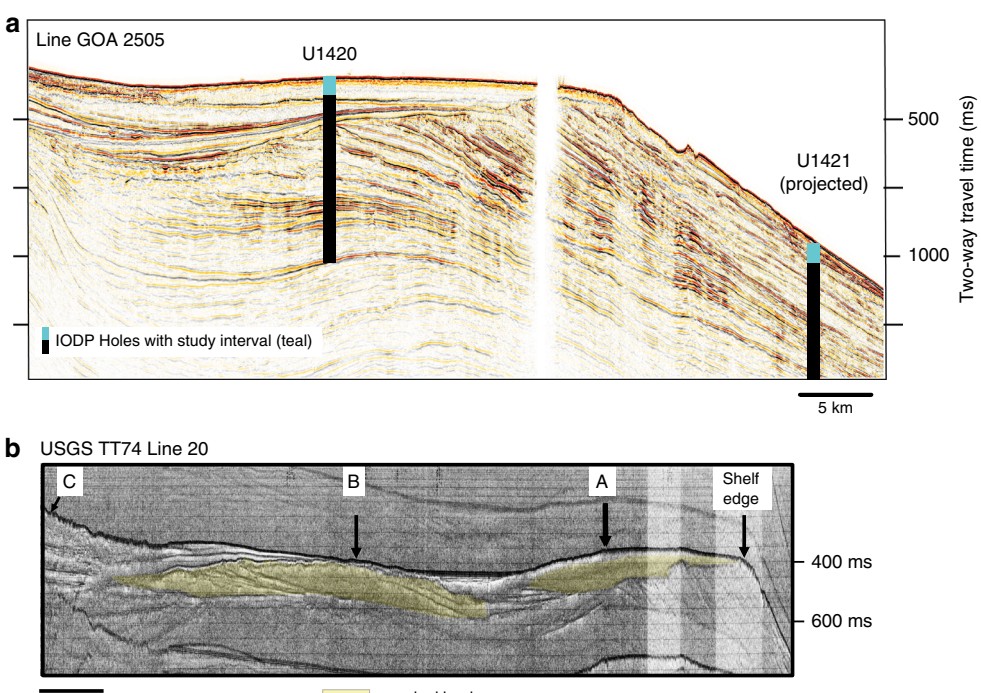

**Fig. 3 Seismic profiles. a** Summary diagram showing position of IODP site U1421 on the slope and site U1420 in the trough along line GOA2505.
**b** Minisparker profile USGS TT74 Line 2025 in Bering Trough illustrating morainal banks corresponding to different terminus positions mentioned in the text. U1421 is located ~4 km east of Line GOA2505.

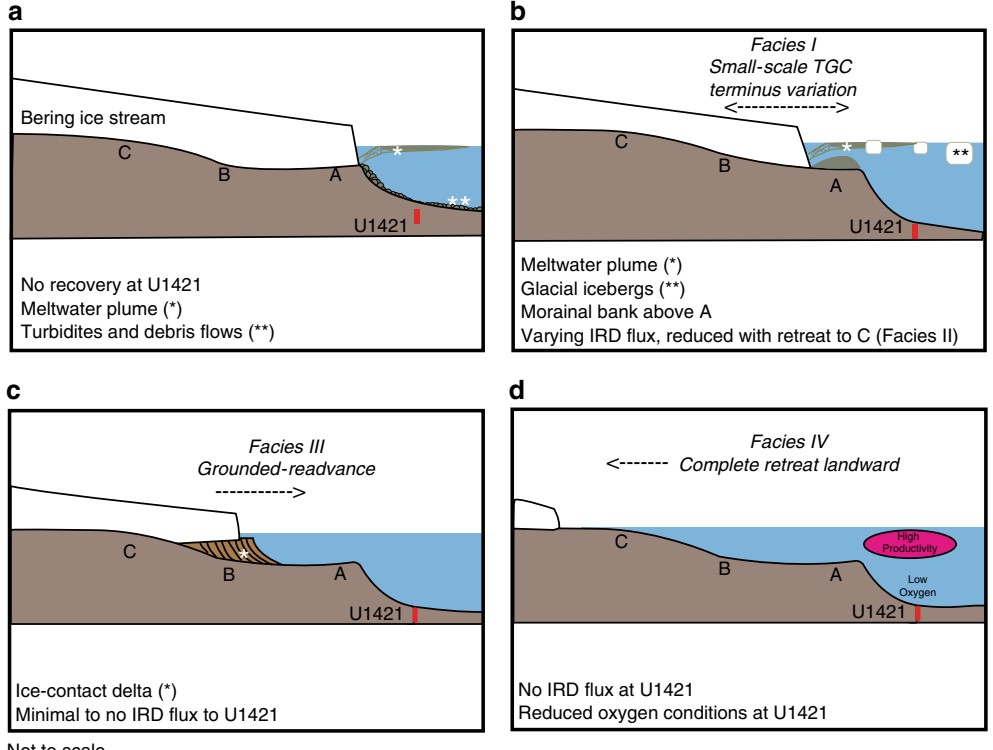

**Fig. 4 Bering ice stream terminus position.** Cartoons showing BIS terminus position within the trough at positions A–C identified on seismic profile in Fig. 3b and paleoenvironmental conditions recorded by sedimentary facies on the continental slope at Site U1421. **a** Maximum advance with terminus at shelf edge. **b** Terminus fluctuates between A and B. **c** Terminus readvance from C to A with formation of ice-contact delta filling inner shelf basin. **d** Complete retreat with terminus landward of C.

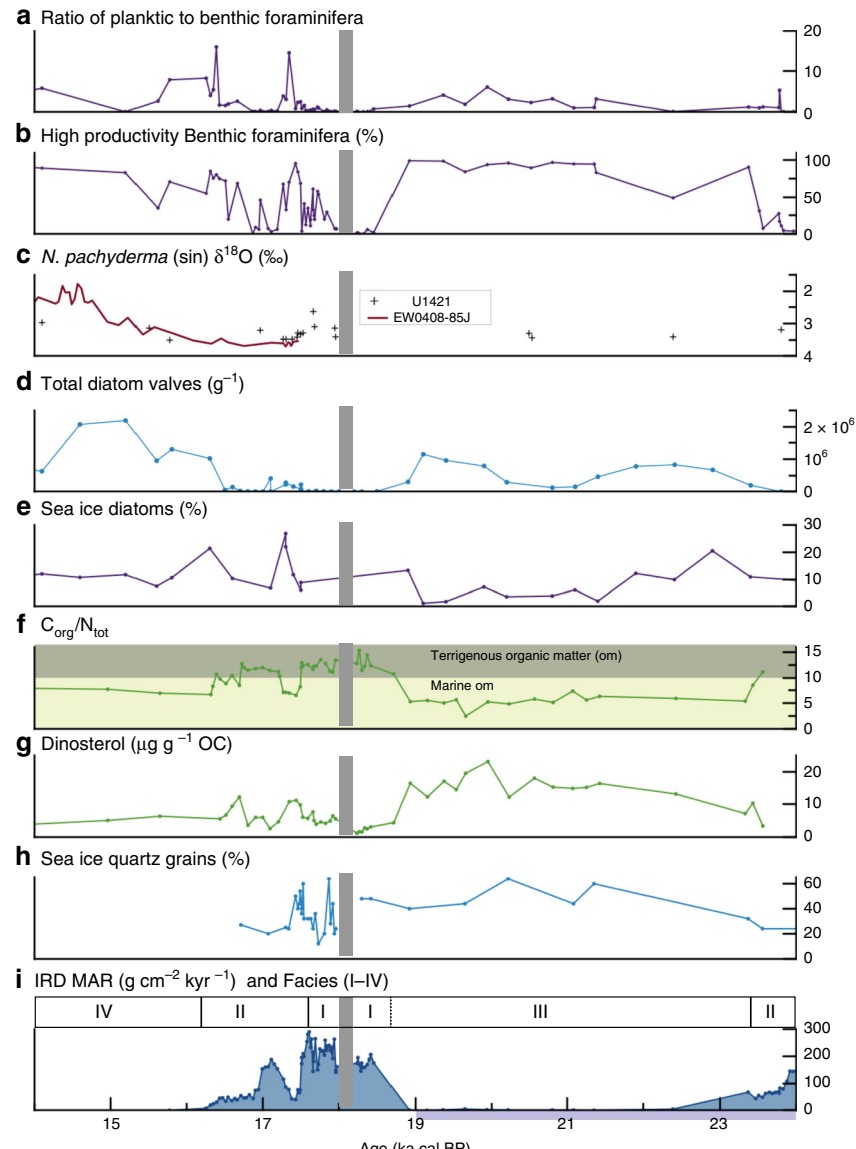

**Fig. 5 Multiproxy records from slope Site U1421 for 24 to 14 kyr ago.** Biotic and biogeochemical conditions at the end of the LGM and during the terminal BIS retreat are shown. Sedimentary records show a sea-ice-dominated glacier margin just prior to terminal retreat. High productivity and a marine source of organic matter occur during periods of low IRD MAR. Quartz microtextures and sea-ice diatoms indicate sea-ice over shelf, reducing IRD MAR at site U1421. **a** Planktic to benthic foraminifera ratio. **b** Percentage of benthic taxa indicative of high productivity and reduced oxygen conditions. **c** Values for $\delta^{18}O$ analysis of *N. pachyderma* (sin.) in samples from site U1421 and survey site EW0408-85JC[27]. Error bars (±1 sigma) are smaller than symbol size. **d** Total diatom valves per gram of sediment. **e** Percentage of sea-ice diatoms. **f** Ratio of organic carbon to total nitrogen. The boundary between marine and terrestrially sourced is drawn at a value of 10[39,40]. **g** Dinosterol ($\mu g g^{-1}$ organic carbon (OC)). **h** Percent quartz grains with sea-ice microtextures. **i** IRD MAR and lithofacies. LGM from 26.5 to 19 ka cal BP[14] is shown by purple bar. Gray shaded area is an interval of no core recovery.

During deposition of Facies I and II, icebergs and meltwater plumes could have extended seaward of the tidewater terminus up to 60 km[34], producing cold, turbid surface waters above the slope (Fig. 4b). Oxygen isotopic analyses, although sparse within the recovered LGM interval, are consistent with results from adjacent continental slope jumbo piston core EW0408-85JC[27] through deglaciation (<17.5 ka cal BP; Supplementary Fig. 5). Isotopically light and variable values that occur earlier may result from the meltwater focused at this slope site near the distal end of the Bering Trough (Fig. 1). Tidewater glacier sedimentary facies have low foraminiferal abundances, low planktic to benthic foraminifera ratios, and few diatoms (Fig. 2d, e). A relatively high diversity assemblage of benthic foraminifera dominated by *Elphidium excavatum sensu lato* and *Cassidulina reniforme*, both

indicators of high meltwater discharge[35], along with *C. teretis*, *Buccella frigida*, *Quinqueloculina* spp., *Islandiella californica*, and *Cibicides* spp. occur in Facies I and II (Supplementary Fig. 6, Supplementary Data 6). When BIS occupied the Bering Trough, shelf environments were nearly eliminated (Fig. 4b) and abundant meltwater shifted water mass properties (e.g., salinity, food availability, etc.) typical of the shelf onto the continental slope. This explains the abundance of *E. excavatum s.l.* and other typical shelf taxa in Facies I and II sediments that show no evidence of sediment gravity flows.

Facies III, deposited from 23.4 to 18.7 ka cal BP, represents the readvancing terminus (shown from C to A on Figs. 3b and 4c) in the TGC. At U1421, this interval consists of mud with abundant to dispersed clasts with an average IRD MAR of

~10 g cm$^{-2}$ kyr$^{-1}$ and lower sedimentation rates between 2 and 6 m kyr$^{-1}$ (Fig. 2f). Sediment is lighter in color than Facies I and II, contains mollusk fragments, a higher planktic to benthic foraminifera ratio (Fig. 5a), lower benthic foraminifera diversity dominated by the low oxygen taxa *Epistominella pacifica* and *Uvigerina peregrina* (Fig. 5b, Supplementary Fig. 6), and abundant diatoms (Fig. 5d). *E. pacifica* and *U. peregrina* are indicators of suboxic conditions along the Northeastern Pacific margin[36] and reduced oxygen may be a factor leading to decreased diversity; however, both the lack of laminations and other low oxygen benthic taxa (Supplementary Fig. 6) suggest that hypoxia was not present at this time. Diatom assemblages are diverse and dominated by *Chaetoceros* resting spores (RS), an upwelling indicator often found along continental slopes in the North Pacific, *Thalassionema nitzschioides*, a diatom associated with high nutrient waters in the North Pacific, and *Shionodiscus trifultus*, a diatom common in regions with melting sea ice. Several diatoms that are associated with sea ice are also present, including *Fragilariopsis cylindrus*, *Thalassiosira antarctica* RS, and *Ehrenbergiulva* spp.[37,38]. Sea-ice diatoms decrease in abundance throughout Facies III (Fig. 5e, Supplementary Fig. 7). Low $C_{org}/N_{tot}$ values[39,40] and elevated concentrations of phytoplankton-sourced dinosterol[41] also support a higher marine productivity. Low bulk density within this interval (Supplementary Fig. 3) results from increased biogenic content and reduced sand (Fig. 2g).

We interpret Facies III as distal to an ice-contact delta on the shelf that trapped sand and separated the grounding line from the sea. Local sea level may have fallen close to the ice stream due to crustal rebound in response to grounding line retreat and weakened gravitational pull of the smaller ice sheet mass on the ocean[42,43], thus aiding BIS advance over the outwash plain as the outer trough basin filled, finally nearing the seaward end of the trough and advancing to the shelf break by 18.2 ka cal BP. The TGC model[13] explains this advance lasting 4500 years (~5.5 m year$^{-1}$) even when regional SSTs were warmer and eustatic sea level was rising after ~21 ka (Fig. 2a, b, Supplementary Fig. 5).

**Terminal retreat**. A calving terminus with high meltwater discharge and turbid surface water near the shelf edge after 17.9 ka cal BP is indicated by high mud and IRD accumulation rates and deposition of Facies I. Low marine productivity occurred at Site U1421, as denoted by low planktic to benthic foraminifera ratio, low diatom abundance as well as reduced dinosterol concentration (Fig. 5a, d, g). Export of terrigenous organic matter delivered by icebergs[44] and meltwater discharge is deduced from elevated $C_{org}/N_{tot}$ ratios (Fig. 5f).

Terminal retreat of BIS from 17.6 to ~16 ka cal BP is characterized by decreasing sedimentation rates of meltwater-derived mud and IRD, with an increasingly productive water column in a warming coastal ocean as eustatic sea level rise was accelerating (Figs. 2, 4c, and 5).

After 17.6 ka cal BP, there is a gradual decrease in IRD MAR that tracks with regional SST increases and low planktic foraminifera $\delta^{18}O$ values (Figs. 2b and 5c, i), suggestive of a warming and/or freshening coastal Gulf of Alaska[27]. This 1000-year-long terminal retreat is interrupted by a prominent drop in IRD between 17.5 and 17.2 ka cal BP (Figs. 2h and 5i). We suggest that during this 300-year period, perennial sea ice on the shelf restricted delivery of IRD to the slope.

The continental slope occurs within the sea-ice-rafting zone and therefore IRD could include grains transported by sea-ice floes as well as by icebergs. We discriminate between these populations by cataloging surface microtextures on quartz grains from scanning electron microscope (SEM) images (Supplementary Fig. 8) based on a study of modern ice floes in the Arctic[45]. Sea-ice-transported grains are rounded and weathered rather than exhibiting sharp-edged mechanically broken textures characteristic of glacial transport (Supplementary Fig. 9). Sea-ice-rafted characteristics occur on 40–60% of the quartz grains from samples with low IRD MAR compared with ≤40% of grains from high IRD MAR samples deposited during the retreat phase of the TGC (Fig. 5h). During the period from 17.5 to 17.2 ka cal BP both the percent of sea-ice-rafted grains and sea-ice related diatoms increased to high levels (Fig. 5e, h, Supplementary Fig. 7). This period coincides with SST minima in the NE Pacific near 17.5 ka[46].

Tidewater glacier advance during sea-ice build up (ice mélange) is well known in Greenland[47–49] and has been documented in front of Tsaa Glacier in Icy Bay, Alaska[1]. The IRD MAR record for BIS seems to show that ice mélange slowed terminus retreat on an open coastline in a similar way as in fjords (Figs. 2 and 5). However, when sea ice broke out at 17.2 ka cal BP releasing icebergs the retreat that began at 17.6 ka cal BP resumed. In contrast to the earlier terminus retreat (from 25.0 to 23.5 ka cal BP), high sedimentation rates from 17.6 to ~16 ka cal BP could not maintain the TGC, and the marine terminus retreated from the Bering Trough. The presence of inland silled fjords would restrict the export of icebergs to the slope after ~16 ka, limiting detailed reconstruction of BIS terminus behavior from the U1421 sedimentary record after this. However, mapping of surficial deposits on the Bering Glacier foreland and recovery of marine invertebrates within these sediments indicate the presence of a rocky fjord coastline, as much as 55 km inland from the present-day shoreline from 16 to 5.5 ka[50]. The Cordilleran ice sheet ice volume modeled using paleotemperature proxy records from Greenland ice cores (GRIP) appear similar to the observed initial retreat from the trough[51].

## Discussion

Outlet glaciers become unstable when sufficient calving initiates retreat at their grounded marine termini. Acceleration of ice flow into the ocean and glacier thinning then propagates upstream into the ice sheet[6]. At the tidewater ice cliff, warm seawater can promote undercutting and collapse, thus accelerating iceberg calving[52] and triggering glacier retreat. Also, sea level rise may increase buoyancy forces on the terminus and alter basin geometry[29,53]. Temperate tidewater glaciers can enhance their stability by building submarine moraincal banks at the terminus, which both reduce buoyancy forces and insulate the ice from warm erosive seawater[3,13]. SST began to steadily rise ~17 ka, exceeding those previously observed in the LGM[46]; however, BIS appears to switch from the TGC and begin its terminal retreat earlier, around 17.6 ka cal BP. This timing suggests that the trigger for terminal retreat may be related to rapid sea level rise, increased subglacial meltwater erosion (Fig. 2c, d), and morainal bank failure. Warming temperatures and increased meltwater discharge should have increased the sediment flux, instead retreat occurred. We envision a scenario where morainal bank construction could not keep pace with rapidly increasing water depth in concert with increasing meltwater discharge accelerating morainal bank erosion in the Bering Trough; together these processes resulted in irrevocable retreat. Along the open shelf, rising sea level inundated the exposed margins of the ice stream increasing the effects of buoyancy and wave action beyond that directed at the downstream end of the trough (Fig. 1). This condition is in contrast to the LGM when the ice stream was governed by the TGC and slowly advanced by building up morainal banks or an ice-contact delta or rapidly retreated when

these stabilizing sediments failed. The tempo of advance and retreat of individual marine termini along the Gulf of Alaska margin of the Cordilleran ice sheet would produce the observed diachronous retreat pattern[54]. Furthermore, a complex pattern of iceberg discharge from the troughs along the Alaskan margin would be expected to produce a variable IRD signal to the NE Pacific that resembled a line source as each ice stream responded to its TGC. In the case of the BIS, retreat into the coastal fjord system eliminated the IRD flux to the slope by 16.3 ka cal BP and decreased sedimentation rates to <1 m kyr$^{-1}$ through the Holocene (Facies IV) (Supplementary Fig. 2).

At IODP Site U1421, muddy lithofacies comprise 86% of the drill core deposited since the onset of the LGM, thus preserving a near-field paleoenvironmental archive within the trough-mouth fan seaward of a major Cordilleran outlet glacier. This paleorecord foreshadows a potential scenario for behavior of polar outlet glaciers in Greenland and Antarctica in a warming world. In these regions, calving ice shelves[55,56] and thawed bed conditions[57] suggest that a polar to temperate transition is already occurring. Glaciers that achieve this threshold release meltwater that taps stored subglacial sediment, increasing sediment flux to the terminus constructing morainal banks at the grounding line that stabilize the marine terminus. We find that this process effectively maintained a Cordilleran marine ice stream near the seaward end of its trough for thousands of years, thus delaying but not forestalling irreversible retreat. Constraints on sediment flux to glacial termini may therefore be critical for defining model parameters for the timescale of individual outlet glacier response to changing climate conditions.

## Methods

**Ice-rafted debris abundance.** Ice-rafted debris is quantified by weighing the coarse sand fraction (250 μm–2 mm) following the method of Krissek[58]. Coarse sand was separated from 10 cm$^3$ samples by wet and dry sieving after air drying and rinsing with distilled water to remove salts. Each sand sample was examined with a binocular microscope to estimate the volume of terrigenous ice-rafted sediment (in volume percent) in order to exclude biogenic components and burrow fills, which do not have an ice-raft origin. MARs were calculated by multiplying by the sedimentation rates and dry bulk densities, which were obtained from discrete shipboard measurements[30] (Supplementary Data 1).

**Age model and sedimentation rates.** Sediment samples from Site U1421 were sieved at 150 μm and picked for benthic and planktic foraminifera, with care taken to avoid infaunal benthic species (as per methods reported for site survey core EW0408-85JC[59]). Radiocarbon analyses were performed at Australian National University at the Single Stage Accelerator Mass Spectrometry (SSAMS) Lab in the Research School of Earth Sciences[60]. Instrument reproducibility was tracked over the course of the project via the analysis of 30 unleached aliquots of the 18,199 ± 8 BP TIRI/FIRI turbidite standard[61]; individual dates averaged 18,210 ± 50 BP (1−σ) and ranged from 18,110 to 18,300 BP with individual reported errors of between 45 and 70 $^{14}$C years.

The age model for the uppermost 300 m CCSF-A at the site was generated by evaluation of the 11 planktic dates available for the core, in addition to one late Holocene benthic radiocarbon date where no planktic data were available via the Bayesian age modeling program BChron[62]. Raw dates were calibrated to the Marine 13 curve[63] assuming a planktic $\Delta R$ of 470 ± 80 (encompassing the range of modern observations[64] after Davies et al.[27], and benthic $\Delta R$ of 1300 ± 100 (based on mid-Holocene $^{14}$C benthic–planktic difference at a similar time and water depth from proximal core site EW0408-85JC[59]). The age model was further constrained via three lithologic transitions at 6.01, 52.078, and 58.078 m CCSF-A, such that $^{14}$C-resolved sedimentation rates within a given depositional unit were extended to its boundaries. Sedimentation rates were then calculated from this age model at binned 1000-year intervals (Supplementary Data 2–4)

**Seismic data.** Single-channel minisparker (400–800 J, 100–900 Hz) profiles were collected by the U.S. Geological Survey in conjunction with the University of Washington in September 1974[65]. Analog records were available on microfilm. We identified the morainal banks within Line 20, collected along the axis of the Bering Trough with guidance from previous work[16].

Seismic line GOA2505 was collected as part of the National Science Foundation, Ocean Drilling Program-funded site survey aboard the R/V Maurice Ewing in the summer of 2004[66]. The sources for the survey were simultaneously firing, dual

45/45 cubic inch GI guns, which are ideal due to their suppression of the bubble pulse. The shot interval was 25 m and the signal was recorded using a 750-m-long, 60 channel hydrophone streamer with a 1 ms sampling interval. Processing steps completed at the University of Texas Institute for Geophysics included: bandpass filtering, normal moveout correction, trace regularization, f-k filtering, water-bottom muting, stacking, and finite-difference migration[66]. Resultant vertical resolution is ~5 m with 1500–2000 m of penetration in sediments.[17,18].

**Oxygen isotopes.** Well-preserved specimens of planktonic foraminifera (*N. pachyderma* (sinistral) and benthic foraminifera (*Uvigerina peregrina* and *Epistominella pacifica*) were picked from the >150 μm size fraction. Isotope analyses were performed by the Oregon State University College of Oceanic and Atmospheric Sciences Stable Isotope Laboratory using a Kiel III carbonate preparation device connected to a Thermo-Finnigan MAT-252 mass spectrometer. Data were corrected using an internal lab standard (Wiley—expected $\delta^{18}O = -7.20$ permil vs VPDB). The international standard NBS19 was run as a check standard (expected $\delta^{18}O = -2.20$ permil vs VPDB). Results were compared with the LR04 benthic stack[67] (Supplementary Fig. 5). Data are presented in Supplementary Data 5.

**Foraminifera assemblages.** A total of 116 samples were examined for foraminifera (Supplementary Data 6). Due to the overwhelming amount of sand in most samples, and the desire to compare samples from other studies of the LGM[35], only the size fraction greater than 150 μm was examined. At least 300 benthic foraminifera were picked, abundance permitting. A microsplitter was used in fossil-rich samples to reduce the amount of sample picked. A split factor was applied to split samples so that samples of different sizes could be compared. All planktonic foraminifera in the picked portion were counted to generate a planktic to benthic ratio. Distribution plots of paleoenvironmentally significant benthic taxa[35,36,68] and Shannon diversity values were generated using PAST 3 software package[69] (Supplementary Fig. 6).

**Diatoms.** A total of 66 samples were processed for diatom analysis (Supplementary Data 7). Sediments were treated to remove organic matter and carbonates[70], then concentrated using the heavy liquid, sodium polytungstate (SPT). SPT was diluted to 2.38 g cm$^{-3}$ and added to the treated sediments. Sediments were centrifuged for 20 min at 2500 rpm. The supernatant, which contained primarily diatoms, was collected and the process repeated twice more. The three vials of supernatant for each sample were then combined into one centrifuge tube, 15 ml of DI water was added to reduce the density of the liquid, and this mixture was centrifuged again for 20 min at 2500 rpm. After decanting and discarding the supernatant, this process was also repeated twice more. The discarded supernatant and discarded dense sediments were checked to be sure no diatoms were present using a Nikon Eclipse Ni compound microscope at ×400 magnification. None was found. The concentrated diatoms were then diluted with sodium hexametaphosphate, a deflocculant, and made into quantitative diatom slides[70]. When possible, at least 300 diatoms were identified to the species level at ×1000 magnification from at least three transects across each slide[71]. In cases with apparent low diatom abundance, diatoms were counted from three transects and the number of diatoms per gram of sediment was calculated using the formula by Scherer[70]. If there were fewer than 25,000 diatoms per gram of sediment, the sample was considered barren and diatoms were not identified to the species level for that sample (Supplementary Fig. 7).

**Organic geochemical analyses.** Bulk sediment total organic carbon ($C_{org}$) content was determined using a carbon–sulfur determinator (ELTRA CS 800) after the removal of carbonates by adding 500 μl 12N hydrochloric acid to the sediment. Total nitrogen ($N_{tot}$) content determined by a CNS analyzer (Elementar III, Vario) was used to calculate atomic C/N ratios to distinguish between algae and land-plant origins of organic matter[72]. For biomarker analyses, sediments were extracted by means of ultrasonication using a dichloromethane:methanol mixture (2:1, v–v). Sediments were extracted three times and the combined total lipid extract was fractionated using silica open-column chromatography to obtain the polar lipids (ethylacetate:*n*-hexane; 20:80, v–v). For quantification, androstanol was added to the sediment as internal standard prior to extraction. Sterols were derivatized using N,O-bis(trimethylsilyl)trifluoroacetamide (BSTFA; 300 μl; 60 °C; 120 min) and analyzed using a gas chromatograph (Agilent 7890B; 30 m DB 1MS column, 0.25 mm diameter, 0.25 μm film thickness) coupled to a mass selective detector (Agilent 5977B; 70 eV). The identification of dinosterol (4α,23,24-trimethyl-5α-cholest-22E-en-3β-ol) was based on comparison of its retention time and mass spectra with that of a reference compound run on the same instrument. Data are presented in Supplementary Data 8.

**Quartz grain microtexture analyses.** Quartz grains from 38 ice-rafted samples were imaged on a JEOL JSM-IT-300LV SEM in the high vacuum mode at 20 kV. Approximately 25 grains from each sample were mounted on aluminum stubs and gold-coated. Elemental (EDS) analysis was used to verify quartz composition before collecting a photomicrograph. We complied a checklist based on the literature that included microtextures typical of glacial crushing and assumed direct iceberg transport including high relief, conchoidal fracture, fracture faces, arcuate steps,

straight steps, and angular (sharp) outlines[73,74] and microtextures identified from grains collected from modern sea-ice floes in the Arctic Ocean. These included medium-low relief, scaling or surface roughness, silica precipitation or solution, and smooth edges[45] (Supplementary Fig. 8). In addition, grains with v-shaped percussion marks indicative of transport by meltwater streams were also noted[45,75]. Images were analyzed for the presence (1) or absence (0) of each microtexture on the list and frequency percent was calculated for each sample. PCA confirms the microtexture associations for our Alaska samples (Supplementary Fig. 9). Within a sample, each grain was assigned to one of the three groups, either sea-ice rafted, iceberg rafted, or mixed (grains with both types of microtextures) and the percent of sea-ice-rafted grains (Supplementary Data 9) was compared with other proxies (Fig. 5).

## Data availability

All data generated or analyzed during this study are included in this published article (and its Supplementary Information and data files).

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

## Acknowledgements

We thank the International Ocean Discovery Program U.S. Implementing Organization, the science party of IODP Expedition 341, and the captain and crew of the D/V JOIDES Resolution. Funding was provided by the National Science Foundation award OCE-1434945 and a post-expedition award from the U.S. Science Support Program of IODP to E.A.C. J.M. received funding from the German Research Foundation (MU3670/1-2) and a Helmholtz Research grant (VH-NG-1101). S.D.Z. received funding from the University of Central Missouri Center for Teaching and Learning. M.H.W. and S.J.F. acknowledge support from the Australian IODP office, Australian Research Council, and American Australian Association. This is the University of Texas Institute for Geophysics Contribution #3644.

## Author contributions

E.A.C. and S.D.Z. conceived the research and wrote the manuscript. E.A.C. analyzed IRD MAR, S.D.Z. analyzed foraminifera, J.M. analyzed biogeochemical proxies, L.L.W., S.P.S.G., and W.A.C. interpreted downhole logs and geophysical data, J.M.J. provided insight into the TGC model and sediment flux, B.E.C. analyzed diatoms, J.W.P. imaged and categorized quartz sand grains, M.H.W. and S.J.F. established the chronology. M.H.W. and A.C.M. interpreted oxygen isotope results. All authors read and commented on the manuscript.

## Competing interests

The authors declare no competing interests.
