## [Peer Review File · Nature Communications]

Reviewers' comments:

Reviewer #1 (Remarks to the Author):

This manuscript uses various measurements from an IODP sediment core, along with seismic profiles, to derive the history of the Bering Ice Stream (BIS), draining from the Cordilleran Ice Sheet, during the Last Glacial Maximum. The authors provide evidence that the sediment dynamics has played an important role in the dynamics of the ice stream. In particular, after the BIS terminus retreated from its maximum extent on the shelf edge, it entered into a cycle of retreat and advance, as the morainal bank collapses and then slowly rebuilds, akin to tidewater glacier cycles seen today. They argue that the cycle breaks and irrevocable retreat ensues when the rate of sea level rise out-paces the rate of sediment deposition. As a result, the work will be of interest to those trying to understand the (in)stability of present-day tidewater glacier systems and feedbacks with sediments, as well as those interested in paleo-glacio-environments.

This manuscript represents an impressive quantity of analysis, which is well presented and documented in the main text, methods, figures and tables. The supplementary data are detailed and allow for others to make their own interpretations. I do not have any substantial comments, although I think there needs to be more discussion about other potential stabilizing feedbacks that can occur when ice sheets retreat (see below).

L13: "During half the LGM" – which half?

L38-39: The connection between this sentence and the rest of the paragraph is not obvious, despite the use of "thus" as a transition. Neither open shelf or fjords have been mentioned previously.

"Evidence for TGC" section: how many retreat-advance cycles are seen in the sediment record? If I look through the text, I can work out what is going on from the timings given, but it would be nice to have the interpretation in a figure (perhaps phases related to Fig 3c could be marked on Fig. 2), as understanding the timing of events is not that easy to follow in the text.

L175-176: If advance linked to sea ice build-up is well known, I think there could be more than one reference. Others could include Todd and Christoffersen, 2014; Howat et al., 2010.

L177-179: what are the possible reasons that the mélange/sea ice did not provide this stabilizing effect in this region? It seems you hint at it with "open water setting", but I think you could be more explicit – is it possibly because the mélange is not constrained by fjord walls, and so doesn't provide the same back pressure / reduction in calving at the CIS termini compared to other glaciers?

L182: "rather than INCREASED SEA SURFACE temperatures" (surely increased melt water production is due to increased [air] temperature?)

L183-190: Could part of this delay between the end of the LGM and the retreat of CIS be due to local sea level effects? While GMSL may have started to rise 19-20 ka ago, the sea level local to CIS could have seen a delayed response, due to gravity and glacial isostatic adjustment – providing a stabilizing effect. E.g. see Gomez et al., 2010, 2015.

L186: How is the abrupt decrease in teleconnection strength relevant?

L228-231: missing closing parenthesis somewhere in this sentence.

Fig. 3c: it might be nice to include the retreat stage as a fourth panel.

Gomez, N. et al. (2010) Sea level as a stabilizing factor for marine-ice-sheet grounding lines. *Nature Geoscience*, 3(12), doi: 10.1038/ngeo1012

Gomez, N. et al. (2015) Sea-level feedback lowers projections of future Antarctic Ice-Sheet mass loss. *Nature Communications*, 6(1), doi: 10.1038/ncomms9798

Howat, I.M. et al. (2010) Seasonal variability in the dynamics of marine-terminating outlet glaciers in Greenland. *Journal of Glaciology*, 58(198), doi: 10.3189/002214310793146232.

Todd, J. and Christoffersen, P. (2015) Are seasonal calving dynamics forced by buttressing from ice mélange or undercutting by melting? Outcomes from full-Stokes simulations of Store Glacier, West Greenland. *The Cryosphere*, 8(6), doi: 10.5194/tc-8-2353-2014.

Reviewer #2 (Remarks to the Author):

Review "Sediment controls dynamic behavior of a Cordilleran Ice Stream at the Last Glacial Maximum by Ellen A. Cowan et al.

The manuscript presents multiproxy record of ice-rafted debris, sedimentation rates, microfossils (foraminifera and diatom assemblages) and geochemistry (oxygen isotopes, TOC) of a marine composite core from IODP site U1421. The IODP site U1421 is located on the continental slope of the Bering Trough, and the core is recovered from 721 m water depth. The studied core is in total 695 meters long, yet this study has focus only on the upper 116 meters. The age control is based on radiocarbon dates on planktic (and one benthic) foraminifera. Based on their results, authors conclude that sedimentation at the Bering-Bagley ice stream terminus controlled the ice stream dynamics during the LGM and deglaciation, and the sedimentation together with ice stream dynamics represents tidewater glacier cycle.

The topic of this research is very interesting, and I believe that the results of this study could potentially be interest of wider glaciology/marine researchers, yet it has serious shortcomings that need to be addressed.

The authors use multiple proxy approach which is a notable strength of this manuscript. Their main method is the grain size analysis, IRD and sedimentation rates, which are all conducted in good resolution, despite the actual number of analyzed samples is not given in the text. However, the supporting analyses based on microfossils and geochemistry are done in a very low resolution (for 116 meters 72 foraminifera samples, 45 diatom samples, TOC not given), leading into a sparse data and sample resolution over 1000 years ($\delta^{18}O$ has >2000 years gap in the record). In several parts of the manuscript the interpretations are based too few datapoints and do not sufficiently support the conclusion. Given this, it is essential that the number of datapoints is increased.

I worry that there is no attempt to correlate results with other marine or terrestrial records from the region. The authors have references to previous work on the TGC, but do not really discuss them or compare. Which brings me to question if TGC really work in the shelf environment, as the previous work are studies from fjord environment. In the continental shelf, the ice terminus is more vulnerable to ocean influence. The ocean's influence on the ice and the local geomorphology, are not considered in this study and their control to the ice dynamics should be discussed more thoroughly.

Another concern is the sedimentation. How are these sedimentation rates in comparison with a sedimentation rates of a modern ice stream? Even given the lower sea level during the LGM, would an ice stream be able to deposit such a high amount of sediment in the deep ocean environment during the cold LGM? What triggers the ice to produce enormous amounts of meltwater during the

time of glacial air temperatures and low solar insolation on the Northern Hemisphere.

The authors should better report data and methods, as it is not sufficient in the current state of the manuscript. A full description of the chronology (including for example a table with dated samples, ^{14}C ages, modelled ages, what species were used and so on) should be presented in the main text and not solely in the supplementary. I counted that there are seven dated depth intervals on the section of interest (ca 16-26 ka), and more dated intervals would certainly be beneficial for this study. From the text I understood that the 695 meters long core represents the last 130 kyr? However, the authors present only the last 50 000 years? Is this because they aimed to present only the part that was dated using the radiocarbon method? If so, how was the rest of the core dated? I would suggest the authors to add supplementary Figure 2 to the main text and perhaps leave out the small inset panel as the figure gets small. The authors have selected the ocean reservoir correction based on studies that encompasses the Mid Holocene and deglacial period (<16000yr), I assume that ΔR should be higher for the LGM. Also, the supplementary data shows some age reversals with added comment that sample labels were switched during washing. If this is correct, such a samples should be removed and not included in the study.

To conclude, as its current stage, I would not recommend publication. The authors present too weak evidence for their conclusions and in some places the results are incorrectly interpreted (see list of detailed points below). The manuscript seems to be in an early stage and structurally unpolished; $\delta^{18}\text{O}$ curve (Fig 2c) is not mentioned anywhere in the text, supplementary figures should be in the order as they are mentioned in the text (now 6 and 7 appear before 5). Use of the term Facies in this context might not be the most accurate as it is not visible as a similar sediment lithology through the core (Facies I and II have different lithological features before and after LGM). Perhaps a phase would be more suitable term? Facies I and II mainly have in common the high number of IRD and relatively higher number of meltwater foraminiferas compared to Facies III. Before LGM Facies I and II consist of diamicton and after LGM of mud, thus these Facies might represent different depositional environments before and after LGM.

To notable improve the manuscript, the authors should strengthen chronology (more dated intervals) and incorporate this in the main text. Correlate their results with existing marine/terrestrial records from this region. Add more datapoints to their additional proxies (as well as add ^{14}C measurements). To be more convincing that sediments control the ice stream dynamics rather than other factors (e.g. ocean), authors should present stronger evidence and bring these factors into discussion.

Lines 68-70: I wonder if these sediment rates can be real, as there is no age control for the parts with no sediment recovery. The sedimentation rates become more robust if you have a chronological control on ending and beginning of each recovered part. Although, as mentioned in the text, this site is very likely to be exposed to downslope transport that could cause age reversal in the chronology. I am also not convinced about that non-recovered parts of the core would be indicative of BIS situated at the continental shelf (study site).

Lines 78-79: Figure 2 shows the opposite, the deposition of mud is associated with lowest sedimentation rates (Facies III).

Lines 89-94: In general, it would be reader friendly to add which time period is discussed to the text, instead of referring to Facies I, etc.

The sedimentological description of Facies I as a clast-rich diamicton holds only in the bottom part of the core and not around 18 kyr when another Facies I occurs. Could it be that this to events no not belong to the same Facies to begin with? Same goes to Facies II, which is described on line 98 as clast-rich to clast-poor diamicton, but around 17 kyr Facies II consist of mud.

Is there any other data to support the assumption on multiple ice stream collapses and advances? Could the lower values come from sea ice holding the ice margin from calving rather than ice advance? What does your sea ice diatom record look for this time period? Here, the interpretation

is based on a single proxy and each collapse is interpreted from a single (or two) datapoint(s).

Lines 100-103: What is the modern SST at the site and how does these values differ from it? When during the studied time period SST was 4-7°C, and was it relevant for the ice margin instability? These values should be placed in a context, and 4-7°C doesn't sound as glacial temperature.

Lines 114: "low planktic to benthic foraminifera ratio (P/B), and few diatoms" I don't understand, based on the figure it is opposite and highest during Facies I and II, whereas the number of diatoms remains the same based on Fig 4e.

Lines 116-120: *Islandiella reniforme* or *Cassidulina reniforme*? Aren't the non-recovered parts of the core evidence for gravity flows, as stated in lines 72-74? Explanation on sea ice and icebergs transporting benthic foraminiferas doesn't sound plausible.

Lines 126-136: Foraminifera abundances are not presented in figure 6. Based on Figure 4a, planktic to benthic foraminifera ratio is not higher during Facies III than Facies II, and actually one could also argue if total number of diatoms is any higher during Facies III than Facies II (16-17 and 25.5 ka). Maybe these suboxic conditions are created by extensive sea ice and poor vertical water movement, as there seems to be sea ice associated diatoms as described in the text.

Lines 148-153: It would be relevant to elaborate more why the highest sedimentation rates of the record could not maintain TGC? Could it be due to increased ocean water temperatures that are driving the ice retreat?

Lines 156-159: Based on Figures 2 and 4, there are no datapoints around 18ka, so what is this based on? Proxies on Figure 4c and 4d are missing data between ca. 17.5 and 19 ka.

Lines 171-174: Would be relevant to show the portion of sea ice rafted IRD during the IRD peaks.

Responses to Reviewers' Comments

Cowan et al., **Sediment controls dynamic behavior of a Cordilleran Ice Stream at the Last Glacial Maximum**

Reviewer #1

Remarks to the Authors	Response (Line numbers refer to revised manuscript without track changes).
This manuscript represents an impressive quantity of analysis, which is well presented and documented in the main text, methods, figures and tables. The supplementary data are detailed and allow for others to make their own interpretations. I do not have any substantial comments, although I think there needs to be more discussion about other potential stabilizing feedbacks that can occur when ice sheets retreat (see below).	Thank you for your constructive comments on the manuscript. We appreciate the opportunity to improve our discussion of the stabilizing feedbacks that can occur in retreating ice sheets.
L13: “During half the LGM” – which half?	Change to “During the latter half of the LGM”
L38-39: The connection between this sentence and the rest of the paragraph is not obvious, despite the use of “thus” as a transition. Neither open shelf or fjords have been mentioned previously.	Change to “Our study demonstrates that the TGC operates in both open shelf and restricted fjord conditions in meltwater-dominated settings.”
“Evidence for TGC” section: how many retreat-advance cycles are seen in the sediment record? If I look through the text, I can work out what is going on from the timings given, but it would be nice to have the interpretation in a figure (perhaps phases related to Fig 3c could be marked on Fig. 2), as understanding the timing of events is not that easy to follow in the text.	We appreciate this suggestion. We have delineated the number of cycles on Supplementary Fig. 4 where the plot is enlarged to show detail and we have referenced this information in our discussion of Facies I in lines 108-109. The duration of each high-frequency cycle is roughly consistent, which is implied by Brinkerhoff et al. modeling. This observation is more significant than the exact time of occurrence of each cycle within Facies I.
175-176: If advance linked to sea ice build-up is well known, I think there could be more than one reference. Others could include Todd and Christoffersen, 2014; Howat et al., 2010.	Thank you for suggesting these appropriate references. We have added them to the manuscript.
L177-179: what are the possible reasons that the mélange/sea ice did not provide	The IRD MAR record indicates that there likely was a stabilizing effect on the

this stabilizing effect in this region? It seems you hint at it with “open water setting”, but I think you could be more explicit – is it possibly because the mélange is not constrained by fjord walls, and so doesn’t provide the same back pressure / reduction in calving at the CIS termini compared to other glaciers?	terminus but only while ice mélange was present. This is consistent with Todd and Christoffersen, 2014. Retreat of the terminus resumed following the break out of thick ice on the shelf.
L182: “rather than INCREASED SEA SURFACE temperatures” (surely increased melt water production is due to increased [air] temperature?)	We are referring here to the sensitivity of the marine terminus to SSTs. Lines 116-121.
L183-190: Could part of this delay between the end of the LGM and the retreat of CIS be due to local sea level effects? While GMSL may have started to rise 19-20 ka ago, the sea level local to CIS could have seen a delayed response, due to gravity and glacial isostatic adjustment – providing a stabilizing effect. e.g. see Gomez et al., 2010, 2015.	Local sea level effects could certainly play a role as would ice load and sediment flux at the terminus. Foraminifer and diatom data (fig. 2) show a pattern that is consistent with IRD until ~16.3 ka. During Facies I (26.7-25.2 ka and 18.7-17.6 ka) we observe small high frequency fluctuations in the terminus position whose duration and distance are well explained by the TGC model and we do not see an obvious shallowing of water depth in these data. However, after retreat to point C (recorded in Facies II) we interpret advance of the grounded terminus in shallow water (buildup of an outwash plain) from ~23-18.5 ka. References to Gomez et al. (2010, 2015) have been added in line 165 to acknowledge that shallowing of relative sea level could be a factor in this advance to the shelf edge.
L186: How is the abrupt decrease in teleconnection strength relevant?	We agree and we have deleted the reference that introduces this topic.
L228-231: missing closing parenthesis somewhere in this sentence.	(assuming a planktic ΔR of 470 ± 80 (encompassing the range of modern observations⁵⁶) after Davies et al.²⁷)
Fig. 3c: it might be nice to include the retreat stage as a fourth panel.	Fourth panel of Figure 3c included showing retreat stage. We have also identified Facies IV, defined by the absence of IRD.

Reviewer #2

Remarks to the Authors	Response (Line numbers refer to revised manuscript without track changes).
The topic of this research is very interesting, and I believe that the results of this study could potentially be interest of wider glaciology/marine researchers, yet it has serious shortcomings that need to be addressed. The authors use multiple proxy approach which is a notable strength of this manuscript. Their main method is the grain size analysis, IRD and sedimentation rates, which are all conducted in good resolution, despite the actual number of analyzed samples is not given in the text.	We appreciate the opportunity to address this review. All sample information is located in supplementary files as opposed to the main text due to space limitations. The number of analyses performed for each proxy is enumerated in the Methods section of the manuscript.
However, the supporting analyses based on microfossils and geochemistry are done in a very low resolution (for 116 meters 72 foraminifera samples, 45 diatom samples, TOC not given), leading into a sparse data and sample resolution over 1000 years ($\delta^{18}O$ has >2000 years gap in the record). In several parts of the manuscript the interpretations are based too few datapoints and do not sufficiently support the conclusion. Given this, it is essential that the number of datapoints is increased.	As the reviewer notes below, accumulation rates at this site are exceptionally high. It's thus more meaningful to consider whether the density of samples over the ~10 kyr period that is the focus of the manuscript are sufficient to support its conclusions. Our approach was to identify descriptive facies that could be used to record the sedimentary signature of the fluctuating ice stream terminus. Microfossils and geochemical analyses provide important supplemental information about in situ bottom conditions and the water column when IRD is deposited. Pursuant to the reviewer's request, we have analyzed 44 additional foraminifer, 21 additional diatom, and 16 additional geochemical samples in addition to clearly indicating barren diatom samples. In addition, we have illustrated the sample depths on Supplementary Fig. 1. These samples filled in gaps in our plots (i.e., Fig. 4), however they did not alter our conclusions.
I worry that there is no attempt to correlate results with other marine or terrestrial records from the region.	All marine records from the region are referenced (these include 18, 25, 26, 27). A review of terrestrial records is included in 52. We have added Yesner et al., 2019 to discuss BIS retreat onshore and Sequinot et al. 2016 to highlight CIS modeling efforts.

The authors have references to previous work on the TGC, but do not really discuss them or compare. Which brings me to question if TGC really work in the shelf environment, as the previous work are studies from fjord environment. In the continental shelf, the ice terminus is more vulnerable to ocean influence. The ocean's influence on the ice and the local geomorphology, are not considered in this study and their control to the ice dynamics should be discussed more thoroughly.	This comment focuses on the purpose of our manuscript. We cannot test the TGC on the modern Alaskan continental shelf because the glaciers have retreated into fjords or further inland. Instead we used seismic profiles and the proximal sediment record to investigate the paleorecord of the TGC at the LGM. We described the morphology of the Bering Trough in lines 49-51 and the extent of BIS is shown in Fig. 1. Our multi-proxy data set shows variable accumulation of IRD MAR that tracks with the terminus fluctuations predicted by the TGC, including slow advance over 3500 yrs to the shelf edge during the 2nd half of the LGM.
Another concern is the sedimentation. How are these sedimentation rates in comparison with a sedimentation rates of a modern ice stream? Even given the lower sea level during the LGM, would an ice stream be able to deposit such a high amount of sediment in the deep ocean environment during the cold LGM? What triggers the ice to produce enormous amounts of meltwater during the time of glacial air temperatures and low solar insolation on the Northern Hemisphere.	The Alaskan Margin has the highest sedimentation rates worldwide as a result of active convergent tectonics and glaciers in a temperate climate. This has been verified over the Quaternary using both marine geophysical data and drill cores and was noted to be particularly intense since the mid-Pleistocene (Gulick et al., 2015) and in the Late Pleistocene (Montelli et al., 2017). Modern sedimentation rates exceed 2 cm/yr on the inner continental shelf (Jaeger et al., 1998). The proximal slope site U1421 is directly in the path of icebergs and suspended sediment plumes from BIS. In the TGC model of Brinkerhoff et al., 2017, sediment composing morainal banks and outwash fans originates as glacifluvial deposits and this is the case in the paleo-sedimentary environment. Episodes of rapid retreat of the CIS documented here are similar in character to the Heinrich Events of the Laurentide and smaller European ice sheets, which occurred during the same period of Earth's history. Much remains unknown about the mechanisms leading to globally distributed observations of millennial-scale instability

	in the ice sheets of MIS 2/3, and speculation on such falls outside the focus of this paper.
The authors should better report data and methods, as it is not sufficient in the current state of the manuscript. A full description of the chronology (including for example a table with dated samples, ¹⁴C ages, modelled ages, what species were used and so on) should be presented in the main text and not solely in the supplementary. I counted that there are seven dated depth intervals on the section of interest (ca 16-26 ka), and more dated intervals would certainly be beneficial for this study. From the text I understood that the 695 meters long core represents the last 130 kyr? However, the authors present only the last 50 000 years? Is this because they aimed to present only the part that was dated using the radiocarbon method? If so, how was the rest of the core dated? I would suggest the authors to add supplementary Figure 2 to the main text and perhaps leave out the small inset panel as the figure gets small. The authors have selected the ocean reservoir correction based on studies that encompasses the Mid Holocene and deglacial period (<16000yr), I assume that ΔR should be higher for the LGM. Also, the supplementary data shows some age reversals with added comment that sample labels were switched during washing. If this is correct, such a samples should be removed and not included in the study.	While our age model is an improvement on the previously published chronology (Montelli et al., 2017; ref 18), the U1421 age model is not the central focus of this paper. Thus, due to the space limitations of the journal format, we feel it is appropriate to provide details of the chronology as Supplementary Fig. 2. With ten ¹⁴C dates between 15-28 ka we have millennial-scale chronological constraint, and an age model 1-σ uncertainty of <500 years for much of the record, expanding to ~700 years between 23-26 ka. While more is always better, the density of dates is sufficient to support the conclusions of this manuscript. Additional dates in the interval between 23-26 ka would have required a very large sample size (as microfossils are diluted by high lithogenic input), and as the additional chronological resolution isn't essential to this study, we could not justify the sample request. Details of the stable-isotope supported chronology for the oldest parts of the stratigraphic record drilled at Site U1421 can be found in Montelli et al., 2017. As described in the text (Lines 65-68), the focus of this manuscript is on the late glacial at Site U1421, where a high degree of stratigraphic recovery (~86%) allows for a construction of a nearly continuous proxy record. The chronology presented here does represent an improvement (via an increased density of dates) on our understanding of sediment accumulation rates in the portion of the U1421 record constrained by radiocarbon. For completeness we thus present the entire age model over the span of calibrated radiocarbon in the

	supplementary tables as well as in Supplementary Fig. 2. We have added clarifying text to the supplement describing the advance of the U1421 ^{14}C record represented by this manuscript. The reservoir correction of ΔR 470 ± 80, reflecting modern pre-bomb oceanographic conditions (McNeely et al., 2006) has been used in a number of published studies on the Gulf of Alaska margin extending back to LGM (e.g. refs. 26, 27, 57). The two samples whose labels were swapped during washing (as demonstrated by re-running a second aliquot picked from the mis-labeled samples, as well as dating of immediately-adjacent intervals requested as replacement from the Gulf Coast Repository) have been removed from Supplement Table 2. They were included in the original submission for the sake of transparency, and do illustrate the high level of analytical reproducibility of the dates (within $\sim 1\%$ for two independently picked aliquots of the same washed sample).
To conclude, as its current stage, I would not recommend publication. The authors present too weak evidence for their conclusions and in some places the results are incorrectly interpreted (see list of detailed points below). The manuscript seems to be in an early stage and structurally unpolished; $\delta^{18}\text{O}$ curve (Fig 2c) is not mentioned anywhere in the text, supplementary figures should be in the order as they are mentioned in the text (now 6 and 7 appear before 5). Use of the term Facies in this context might not be the most accurate as it is not visible	We appreciate the opportunity to improve the manuscript during revision. In the original text the U1421 planktic $\delta^{18}\text{O}$ record was interpreted with regard to changes in oceanographic conditions during terminal BIS retreat (Lines 184-186). We now also introduce broader regional context for the $\delta^{18}\text{O}$ data (Fig 2c) earlier in the manuscript (Lines 124-128). The introduction of Supplementary figures has been rearranged to match the order in the manuscript. We have chosen to use the term “Facies” in the descriptive sense, as is done in

as a similar sediment lithology through the core (Facies I and II have different lithological features before and after LGM). Perhaps a phase would be more suitable term? Facies I and II mainly have in common the high number of IRD and relatively higher number of meltwater foraminiferas compared to Facies III. Before LGM Facies I and II consist of diamicton and after LGM of mud, thus these Facies might represent different depositional environments before and after LGM. To notable improve the manuscript, the authors should strengthen chronology (more dated intervals) and incorporate this in the main text. Correlate their results with existing marine/terrestrial records from this region. Add more datapoints to their additional proxies (as well as add ^{14}C measurements). To be more convincing that sediments control the ice stream dynamics rather than other factors (e.g. ocean), authors should present stronger	sedimentary and seismic stratigraphy literature, to refer to stratigraphic units that exhibit characteristics that differ in grain size, sedimentation rates, macro- and microfossils, and organic chemistry. Ultimately these facies are designated with an environmental interpretation that can be used to infer activity of the BIS terminus (i.e., Facies I – short period TGC cycles, Facies II – sustained retreat within the trough, Facies III – slow advance with the terminus separated from the sea by an outwash plain). The major differences between the lithologies described from the iceberg-influenced section of core pertain to the clast abundance, which is estimated from the split core surface. We assert that the processes of deposition are the same in clast-poor diamicton and mud with dispersed clasts, although the number of pebbles differs (for any number of reasons). Per the suggestion of Reviewer 1 we have introduced Facies IV in this revision to mark the appearance of mud without coarse sand and granules. In addition, the sedimentation rate is low and the foraminifera are indicative of dysoxic conditions (e.g. buliminids and bolivinids). This facies is deposited after BIS has grounded on shore within the coastal mountains and no longer discharges icebergs to Site U1421. The main goal of the manuscript was to illustrate the cyclic behavior of marine-terminating temperate ice streams rather than presenting a chronology of BIS. This point was addressed above. We have added additional data to strengthen our arguments; however as discussed above we are not able to increase the ^{14}C measurements. We again note that
---	---

evidence and bring these factors into discussion.	the present density of dates (~1 per 1000 years) is sufficient to support the conclusions of this study.
Lines 68-70: I wonder if these sediment rates can be real, as there is no age control for the parts with no sediment recovery. The sedimentation rates become more robust if you have a chronological control on ending and beginning of each recovered part. Although, as mentioned in the text, this site is very likely to be exposed to downslope transport that could cause age reversal in the chronology. I am also not convinced about that non-recovered parts of the core would be indicative of BIS situated at the continental shelf (study site).	In Line 65 the long-term accumulation rates for the site are cited in published, peer-reviewed literature (Montelli et al., 2017). It is also worth noting the exceptionally high sedimentation rates observed at Site U1421 are wholly consistent with other Alaskan margin sites drilled during Expedition 341 (e.g. Gulick et al., 2016). Assuming sedimentation reflects suspension settling, rapid accumulation rates are advantageous to high-resolution ¹⁴C dating as well as other biogenic proxy reconstructions, as the lithogenic flux physically dilates the temporal resolution of the microfossil record. As a result, age reversals are unlikely to be observed in the ¹⁴C record, and none of the 26 dates now available for the record show any such reversal. For well-recovered fine-grained sediment we have demonstrated a consistent pattern of IRD, microfossils and biogeochemical proxies that supports settling from suspension rather than by sediment gravity flows. In these high accumulation rate environments, it is unlikely that bioturbation would erase the physical structures (such as turbidites) associated with down-slope transport, which were not identified in the sampled intervals of the recovered cores as noted in the text (Line 133-136). With regard to the comment on the likelihood of down-slope transport, the reviewer is likely referring to our interpretation of the facies which could not be recovered via the APC system on the JOIDES Resolution, discussed in greater detail below. Cryptic down-slope transport (such as in fine-grained turbidite tails) in the recovered portion of the stratigraphic record is a

	possibility. This presumably would result in a preferential impact on the radiocarbon age of benthic species that have a depth tolerance allowing them to reside on the continental shelf as well as on the slope versus those species with a strong preference for slope habitats. Radiocarbon dating of samples for which benthic foraminiferal assemblages were separated by their depth preference in the modern environment indicates no such offset (Table 2), and we have added brief text clarifying these results and their implication to the manuscript (Line 139-141). With regard to the final comment: as stated in the text, the non-recovered portions occur in stratigraphic intervals that were not recovered in multiple attempts despite drilling several holes. The core-log-seismic integration (Supplementary Fig. 3) illustrates the similarity between the 2 non-recovered intervals and also their differences from the recovered part of the record. The tie between the non-recovered intervals and chaotic to semi-transparent units within seismic reflection profiles strongly suggests downslope transport in these intervals. Given the context of the surrounding proxy record suggesting a marine terminating ice sheet in an advanced position on the continental shelf, we thus suggest a mechanism of gravity flow initiation triggered by the arrival of BIS at the shelf edge (similar to the Disko Trough-Mouth Fan described in Marine Geology v. 402 by Ó Cofaigh et al., 2018). We have added clarification on these points to the text (Lines 69-80).
Lines 78-79: Figure 2 shows the opposite, the deposition of mud is associated with lowest sedimentation rates (Facies III).	Added: “and only fine sediment was transported to site U1421”.
Lines 89-94: In general, it would be reader friendly to add which time period is discussed to the text, instead of referring to	As stated, we have organized this manuscript around the cyclic behavior of marine-terminating temperate ice streams

Facies I, etc. The sedimentological description of Facies I as a clast-rich diamicton holds only in the bottom part of the core and not around 18 kyr when another Facies I occurs. Could it be that this to events no not belong to the same Facies to begin with? Same goes to Facies II, which is described on line 98 as clast-rich to clast-poor diamicton, but around 17 kyr Facies II consist of mud. Is there any other data to support the assumption on multiple ice stream collapses and advances? Could the lower values come from sea ice holding the ice margin from calving rather than ice advance? What does your sea ice diatom record look for this time period? Here, the interpretation is based on a single proxy and each collapse is interpreted from a single (or two) datapoint(s).	rather than the history of BIS. Therefore we have identified the facies at the slope Site U1421 to denote specific behavior of the BIS terminus on the shelf. We have annotated the reoccurring pattern in Supplementary Fig. 4 to better support the reoccurring ice stream collapses and advances. Although the number of clasts is variable and this characteristic is used to name the lithology, we define the IRD MAR using the coarse sand fraction, which is better represented within IODP drill cores. As predicted by the model, the collapse phase is typically represented by fewer data points (shorter) than the advance phase (longer). We also observe differences between the TGC (Facies I) and sea ice buildup between 17.5-17.2 ka in diatoms, foraminifera, Corg/Ntot, and Dinosterol as well as in the IRD population (see Fig. 4).
Lines 100-103: What is the modern SST at the site and how does these values differ from it? When during the studied time period SST was 4-7°C, and was it relevant for the ice margin instability? These values should be placed in a context, and 4-7°C doesn't sound as glacial temperature.	Modern SST on the slope is 14°C (ref:27). Tidewater glaciers are sensitive to SSTs (ref: 33). For completeness we include the only records that exist from the region during our study interval in Fig. 2b to show variability in SSTs. We tried but were unable to generate a geochemical-based STT record from site U1421 probably due to the extremely high sedimentation rates in this ice proximal location.
Lines 114: “low planktic to benthic foraminifera ratio (P/B), and few diatoms” I don't understand, based on the figure it is opposite and highest during Facies I and II, whereas the number of diatoms remains the same based on Fig 4e.	Replaced with (see lines 148-151): Sediment is lighter in color than Facies I and II, contains mollusk fragments, a higher planktic to benthic foraminifera ratio (P/B) (Fig. 4a), lower benthic foraminifera diversity dominated by the low oxygen taxa Epistominella pacifica and Uvigerina peregrina (Fig 4b, Supplementary Fig. 6), and an increased diatom abundance (Fig. 4f). E. pacifica and U. peregrina are indicators of suboxic conditions along the Northeastern Pacific margin³⁶ and reduced oxygen may be a factor leading to decreased diversity; however, both

	the lack of laminations and other low oxygen benthic taxa (Supplementary Figure 6) suggest that hypoxia was not present at this time.
Lines 116-120: Islandiella reniforme or Cassidulina reniforme ? Aren't the non-recovered parts of the core evidence for gravity flows, as stated in lines 72-74? Explanation on sea ice and icebergs transporting benthic foraminiferas doesn't sound plausible.	A mistake was made. Yes, it should be Cassidulina reniforme .
Lines 126-136: Foraminifera abundances are not presented in figure 6. Based on Figure 4a, planktic to benthic foraminifera ratio is not higher during Facies III than Facies II, and actually one could also argue if total number of diatoms is any higher during Facies III than Facies II (16-17 and 25.5 ka). Maybe these suboxic conditions are created by extensive sea ice and poor vertical water movement, as there seems to be sea ice associated diatoms as described in the text.	Foraminifera abundances (# of specimens) are presented in Supplementary Figure 6. We incorrectly labeled panel 4h between 14 and ~16 ka. We have now defined this as Facies IV. Yes, this is one possibility.
Lines 148-153: It would be relevant to elaborate more why the highest sedimentation rates of the record could not maintain TGC? Could it be due to increased ocean water temperatures that are driving the ice retreat?	This is an interesting question that addresses the effect of tipping points within the TGC. Clearly sediment flux cannot fight sea level rise forever but it has a delaying action (up to kyr timescale). We can only speculate as to the final tipping point despite the continued high sediment flux but these include elevated SSTs, loss of a protective ice mélange, as well as accelerating sea level rise.
Lines 156-159: Based on Figures 2 and 4, there are no datapoints around 18ka, so what is this based on? Proxies on Figure 4c and 4d are missing data between ca. 17.5 and 19 ka.	We have revised the timing to refer to the period after 17.9 ka when BIS pulled back from the shelf edge. We have modified the text to refer only to Facies I. We have added data points to Figure 4, a, b, d, e, f, g.
Lines 171-174: Would be relevant to show the portion of sea ice rafted IRD during the IRD peaks.	The quartz grain microtexture analysis is an informative yet qualitative measure to document the presence of sea ice transport. We do not believe that it is appropriate to use it to prorate the IRD MAR peaks.

REVIEWERS' COMMENTS:

Reviewer #1 (Remarks to the Author):

I am satisfied the authors have addressed the comments I have made, as well as those of the other referee (insofar that I understand the methodological concerns they raised).

I just have one minor follow up point:

L167: "Local sea level may have shallowed due to crustal rebound..." this is one component of the local sea level feedback I brought up in my first review. Another (described in Gomez et al., 2010, 2015) is the weakened gravitational pull of the reduced ice sheet mass on the ocean, resulting in sea level fall close to the ice sheet. Perhaps this could be added.

Reviewer #2 (Remarks to the Author):

The authors have made some improvement on the manuscript, yet few small details remain. I am delighted to see that the authors have added more datapoints as this issue was one of the major weakness of the earlier manuscript. I feel that the interpretation of the results is now more solid and adds value to research.

Despite having now slightly more discussion of other factors controlling ice sheet dynamics, the authors tend to present sediments as the only driver controlling ice dynamics in their conclusions, yet this is more likely to be combination of different factors. This issue becomes very clear when the highest sedimentation rates are not able to maintain TGC. Having honest discussion of the other controlling factors, and presenting sediments as one of the controlling factor, is not off from the value of this paper.

Lines 15-17: "sediment dynamics among other controlling factors can control"

Line 35: at the glacier terminus

Lines 38-39: This study doesn't provide evidence for TGC in the fjord conditions, please add references or modify sentence.

Line 79: Please specify "other proxy indicators".

Line 86: "when grounding line retreated onto land" – did ice margin become land terminating already at this point?

Line 117: Sutherland et al. discusses sub surface water influence for tidewater glaciers in fjord setting. This is not a reference for SST influence.

Would it be possible to add the locations of MD02-2496 and EW0408-85JC to Fig 1?

Is there evidence of warmer water microfossil taxa in the studied core that could be linked to warmer SSTs on your study site, and thus strengthen your theory?

Line 130: Check reference to the figure.

Lines 133-136: As stressed earlier – Aren't the non-recovered parts of the core are evidence of gravity flow in your core? Sea ice and icebergs are not be able to transport benthic foraminifera in these depths. Please, delete this sentence as it makes no sense.

Lines 132: "other rare taxa" – What species are these?

Line 152: Check reference to the figure.

Line 180: Check reference to the figure.

Line 185-186: Based on the figure neither IRD MAR or C/N-ratios are decreasing between 17.6 and 16.6 cal ka BP; there is peak in IRD and small increase in the C/N-ratios. Here you talk about increased diatom abundances (that are not really increasing during this time period) yet referring to figure 4e (sea ice diatoms).

Line 187: Lower > Low

Lines 189-191: Perennial sea ice – how does perennial sea ice at the site correlate with the highest sedimentation rates of your record? Would perennial sea ice allow such a high deposition of sediments? Yet similar sea ice conditions are characterizing Facies III where the record show continuous high levels of sea ice quartz grains (and lower sedimentation rates that fit perennial/high sea ice concentrations).

218-220: This sentence doesn't make sense. Sea level rise and increased meltwater production are due to temperature increase and increase in the Northern Hemisphere solar insolation, so temperature is driving it.

Fig 2: Add the location of the MD02-2496 and EW0408-85JC to Fig 1.

Fig: 4 f) Vascular land plants have C/N-ratio over 20.

c) d) and i) Is it necessary to show these here as they are presented in Fig 2?

Responses to Reviewers' Comments (Final revisions)

Cowan et al., **Sediment controls dynamic behavior of a Cordilleran Ice Stream at the Last Glacial Maximum**

Reviewer #1

Remarks to the Authors	Response
I am satisfied the authors have addressed the comments I have made, as well as those of the other referee (insofar that I understand the methodological concerns they raised).	Thank you for your careful review of the manuscript.
L167: "Local sea level may have shallowed due to crustal rebound..." this is one component of the local sea level feedback I brought up in my first review. Another (described in Gomez et al., 2010, 2015) is the weakened gravitational pull of the reduced ice sheet mass on the ocean, resulting in sea level fall close to the ice sheet. Perhaps this could be added.	We have added the suggested sentence for clarification to the previous additions.

Reviewer #2

Remarks to the Authors	Response
The authors have made some improvement on the manuscript, yet few small details remain. I am delighted to see that the authors have added more datapoints as this issue was one of the major weakness of the earlier manuscript. I feel that the interpretation of the results is now more solid and adds value to research. Despite having now slightly more discussion of other factors controlling ice sheet dynamics, the authors tend to present sediments as the only driver controlling ice dynamics in their conclusions, yet this is more likely to be combination of different factors. This issue becomes very clear when the highest sedimentation rates are not able to maintain TGC. Having honest discussion of the other controlling factors, and presenting sediments as one of the controlling factor, is not off from the value of this paper.	We have taken this opportunity to expand our discussion section to include variables that can also initiate outlet glacier retreat (including adding additional references for clarification).
Lines 15-17: "sediment dynamics among other controlling factors can control"	We have not modified this sentence in the abstract because we have concluded that our results support the TGC as modelled by Brinkerhoff et al. 2017 and we wish to highlight this here. As stated above, we have

	highlighted the other possible controls in the discussion section.
Line 35: at the glacier terminus	Modified as suggested.
Lines 38-39: This study doesn't provide evidence for TGC in the fjord conditions, please add references or modify sentence.	We have modified the sentence because we are referring to the definition of the LGM by citing this reference not the TGC.
Line 79: Please specify "other proxy indicators".	We have eliminated this statement.
Line 86: "when grounding line retreated onto land" – did ice margin become land terminating already at this point?	Yes, we are referring to 16 kyr. We have added reference to panel d in Figure 4 (formerly part of Figure 3).
Line 117: Sutherland et al. discusses sub surface water influence for tidewater glaciers in fjord setting. This is not a reference for SST influence. Would it be possible to add the locations of MD02-2496 and EW0408-85JC to Fig 1? Is there evidence of warmer water microfossil taxa in the studied core that could be linked to warmer SSTs on your study site, and thus strengthen your theory?	This reference refers to the sensitivity of the ice margin to temperature. We think this is appropriate here. The locations of both core sites are added to Fig. 1. MD02-2496 is shown on the inset map since it is located off of British Columbia. Warm water microfossils (diatoms and planktic foraminifera) are rare and sparsely distributed and nothing conclusive can be stated regarding sea surface temperatures.
Line 130: Check reference to the figure.	Corrected. Thank you.
Lines 133-136: As stressed earlier – Aren't the non-recovered parts of the core are evidence of gravity flow in your core? Sea ice and icebergs are not be able to transport benthic foraminifera in these depths. Please, delete this sentence as it makes no sense.	Sentence has been removed.
Lines 132: "other rare taxa" – What species are these?	Names of rare taxa have been added. This section has been revised and reference to Supplementary Data (Foram) counts has been added.
Line 152: Check reference to the figure.	Corrected. Thank you.
Line 180: Check reference to the figure.	Corrected. Thank you.
Line 185-186: Based on the figure neither IRD MAR or C/N-ratios are decreasing between 17.6 and 16.6 cal ka BP; there is peak in IRD and small increase in the C/N-ratios. Here you talk about increased diatom abundances (that are not really increasing during this time period) yet referring to figure 4e (sea ice diatoms).	We agree that this sentence is too general as written. We have modified it to refer to the decrease in IRD MAR indicating terminal retreat and have deleted the rest.

Line 187: Lower > Low	Made modification.
Lines 189-191: Perennial sea ice – how does perennial sea ice at the site correlate with the highest sedimentation rates of your record? Would perennial sea ice allow such a high deposition of sediments? Yet similar sea ice conditions are characterizing Facies III where the record show continuous high levels of sea ice quartz grains (and lower sedimentation rates that fit perennial/high sea ice concentrations).	The highest sedimentation rates generally correspond with BIS at the shelf edge and Facies I of the TGC. The period with perennial sea ice is environmentally significant (sea ice does not occur along the open coast today) but short (lasting 300 years). The sedimentation rates remain elevated because they are averaged over 1000 years. Facies III has low sedimentation rates because sediment is deposited within the Bering Trough rather than on the slope. IRD is very low because BIS does not have a tidewater terminus at this time. Therefore, the portion of grains rafted by sea ice appears greater.
218-220: This sentence doesn't make sense. Sea level rise and increased meltwater production are due to temperature increase and increase in the Northern Hemisphere solar insolation, so temperature is driving it.	We have rewritten the discussion to consider the controlling variables on outlet glacier terminus position. We have deleted this statement and the reference to it.
Fig 2: Add the location of the MD02-2496 and EW0408-85JC to Fig 1.	They have been added.
Fig: 4 f) Vascular land plants have C/N-ratio over 20. c) d) and i) Is it necessary to show these here as they are presented in Fig 2?	The sediments that we analyzed consist of a mixture of marine and terrigenous derived organic matter (not pure vascular or algae tissues). Setting the C/N threshold to a value of ten is quite common as plankton mainly exhibit C/N ratios of less than 10. We refer to two references to document this (Refs 39, 40 and the references therein). We have included the graphs in Fig. 5 for reference so they can be compared with the other data which is shown at a higher resolution scale.